# Why Tic Severity Changes from Then to Now and from Here to There

**DOI:** 10.3390/jcm11195930

**Published:** 2022-10-08

**Authors:** Ann M. Iverson, Kevin J. Black

**Affiliations:** 1School of Medicine, Washington University in St. Louis, St. Louis, MO 63110, USA; 2Departments of Psychiatry, Neurology, Radiology and Neuroscience, Washington University in St. Louis, St. Louis, MO 63110-1010, USA

**Keywords:** Tourette Syndrome, tic disorders, tic, environment, severity, fluctuations

## Abstract

Much of the research regarding Tourette’s syndrome (TS) has focused on why certain individuals develop tics while others do not. However, a separate line of research focuses on the momentary influences that cause tics to increase or decrease in patients who are already known to have TS or another chronic tic disorder (CTD). Environmental and internal variables such as fatigue, anxiety, and certain types of thoughts all have been shown to worsen tic severity and may even overcome the positive effects of treatment. Other influences such as stress, distraction, and being observed have had mixed effects in the various studies that have examined them. Still, other variables such as social media exposure and dietary habits have received only minimal research attention and would benefit from additional study. Understanding the impact of these environmental and internal influences provides an opportunity to improve behavioral treatments for TS/CTD and to improve the lives of those living with these conditions. This review will examine the current literature on how these moment-to-moment influences impact tic expression in those with TS/CTD.

## 1. Tics Change

Tourette Syndrome (TS) affects approximately 0.5% of the pediatric population in the United States and can be highly debilitating for patients [1]. Although TS is chronic, tic severity and frequency are far from constant over time. Changes in tics are so typical that they have been part of the definition of TS. The Tourette Syndrome Classification Study Group criteria for definite TS required that “the anatomic location, number, frequency, complexity, type, or severity of tics changes over time.” [2] Changes in tics are also incorporated into the Diagnostic Confidence Index. The following criteria make up 15% of the DCI: “waxing and waning course,” “some tics have disappeared, some new tics have appeared,” “environment dependent (not only during stress),” “tics voluntarily suppressible,” and “tics are suggestible” [3].

Much of the research on TS focuses on why some patients develop tics and why others do not, including genetic and environmental factors. However, somewhat less research has focused on what causes tics at any given time point for someone who already has tics. For example, does stress increase tics? If so, why? As Himle et al. put it, “tic variation may in some cases reflect context-dependent interactive learning processes such that once tics [develop], they are shaped and influenced by environmental contingencies.” [4] Many patients view the circumstances in which TS symptoms transiently worsen or improve as very important [5]. This topic was covered in a review by Conelea et al. in 2008, but since that time, new research has continued to add to our understanding of how moment-to-moment factors affect tics [6].

One way to think of how such factors modify tic severity is the ABC model of behavior, which examines antecedents and consequences of any behavior (action), both of which shape how likely that behavior is to occur. For example, if a child hits his sibling, an antecedent might be that the sibling stole his toy, and a consequence might be his parent putting him in time-out. Both the antecedents and the consequences can influence the likelihood that the behavior will be repeated. Different children may have different baseline characteristics that impact their response to particular antecedents and consequences, such as early childhood experiences, anxiety disorders, ADHD, genetic burden, and so on. Even so, Himle et al. found that “tic-exacerbating antecedents and consequences were nearly ubiquitous in a sample of children with chronic tic disorder” [4].

Understanding these environmental impacts on tics can inform both treatment and parenting strategies. For example, understanding how parental reactions to tics reinforce tics can help parents understand how to best support their children. Additionally, understanding how stress or major life events can trigger increased tic activity may help clinicians understand in what circumstances treatment alterations may be necessary. Finally, understanding the broader context in which their child’s tics occur can help families know when to expect changes in tics. For instance, much research suggests a worsening in tics is common in patients ages 8–10 [7], possibly due to “more tic-exacerbating consequences” [4]. This knowledge can aid clinicians in giving anticipatory guidance to parents.

## 2. Situations That Make Tics Better and Worse: What We Think We Know

According to one survey, the most common factors that patients with TS and their families associated with worsened tics include stress or anxiety, fatigue, holidays, birthdays, and return to school in the fall [8]. Some patients report that tics become worse in scenarios where they are particularly self-conscious of their tics, noting that “the desire to suppress them tends to make them much worse,” or “my tics get worse in any situation w[h]ere [I] particularly do not want them noticed” [9]. Additionally, some patients report that any situation in which their tics are acknowledged increases their tics: “talking about my tics … invariably makes me tic much more” [9]. Patients also reported on situations where they felt their tics were improved, such as being “deep in concentration with something else that [they are] doing” [9].

However, there is reason to believe that patient self-report measures are not always accurate. In one 2016 study, children were videotaped and surveyed about their tics during five different situations [10]. Self-reports and videotaped findings were not well correlated, but participants with higher premonitory urges to tic had more accurate self-reports. Additionally, a study that developed a scale to assess the environmental consequences for ticcing found that child-reported data may be more valid than parent-reported data [11]. These concerns regarding the validity of different types of data are important to keep in mind when evaluating the literature regarding tic severity in different situations. Next, we assess to what extent the data support the perceptions outlined above.

## 3. “Just the Facts, Ma’am.” What’s the Evidence about What Situations Makes Tics Better and Worse?

### 3.1. Sleep

A majority of patients feel that fatigue worsens their symptoms [8,12]. In a polysomnography-based study, tic severity was positively correlated with the number of nighttime awakenings and negatively correlated with sleep efficiency, suggesting that sleep loss worsens tics [13]. Additionally, the study found that patients with TS have significantly worsened sleep efficiency than controls [13]. The surveys also show that many patients with TS exhibit parasomnias such as sleep walking (33%), sleep talking (60%), or night terrors (26%) [14]. Given that sleep problems are common in TS, they represent an attractive target for treatment, and indeed there are case reports of tics improving after sleep problems improve [15]. It is possible that sleep loss and fatigue worsen tics primarily due to a decreased ability to manage other tic triggers when fatigued [16].

### 3.2. Anxiety

Anxiety is another commonly cited trigger for tics. In one survey of 14 children with TS, the most common noted exacerbating factor was “being upset or anxious” [12]. A more recent study from Himle et al. looked at antecedents and consequences of tics using the function-based assessment of tics (FBAT) [4]. In this study, children diagnosed with comorbid internalizing disorders such as anxiety reported more tic-exacerbating antecedents than those without comorbid internalizing disorders. Additionally, the total number of antecedents was correlated with panic and somatic anxiety, and the total number of consequences was correlated with school anxiety and avoidance [4]. It is possible that children with anxiety disorders have a lower threshold for becoming anxious in response to other triggers, therefore leading to increased tics in response to a greater number of triggers. It is also possible that anxious children are more sensitive to others’ reactions to their tics, thus leading to more perceived consequences for their tics [4].

A later study of 45 children with TS by Eaton et al. showed that children with higher separation anxiety symptoms had more tics and that children with higher internalizing symptoms experienced more environmental consequences for tics. However, in this study, the environmental consequences themselves accounted for more of the variance in tics than the internalizing symptoms [17]. This suggests that while anxiety mediates the environmental impact of tics, anxiety alone does not explain differences in tic severity.

Some patients with TS experience so-called “tic attacks,” which are prolonged (>15 min) clusters of tics or tic-like movements. In one case series of 12 children with tic attacks, all children reported comorbid anxiety symptoms [18]. During the attacks themselves, they reported increased internal focus of attention on tics, suggesting a vicious cycle of anxiety leading to worsened tics and vice versa. In a different study of 75 children with tic disorders and their parents, increased family accommodation was associated with increased anxiety, as well as many other factors, including increased depressive symptoms, externalizing symptoms, aggression, rule-breaking behaviors, and functional impairment from tics [19]. Interestingly, despite being associated with many negative outcomes, increased family accommodation was not associated with increased tic severity.

However, as Godar and Bortolato [16] explained, the relationship between anxiety and TS is not clear. Several studies show a greater physiological response to stressors in patients with TS [20,21,22]. However, a recent study found that patients with TS actually showed lower levels of evening cortisol than controls [22], suggesting that perhaps tics reduce anxiety for patients with TS. Alternatively, it is also possible that the lower evening cortisol markers are simply a marker of chronic stress, which has been shown in multiple studies in non-TS patients [22,23,24,25,26].

### 3.3. Thoughts

Specific types of thoughts have also been associated with increased tics. In one study, 60 patients with TS/CTD were surveyed on their thoughts preceding tics, including three categories: anticipation (that tics may occur), interference (that they may negatively impact activities, self, or appearance), and permission (whether or not the person felt they were allowed to tic) [27]. The participants reported that all three categories of thoughts led to tics. Importantly, the interference and anticipation subscales of the Thinking About Tics inventory (THAT) were reduced in participants after cognitive behavioral therapy (CBT). Godar and Bortolato suggest that perhaps boredom leads to an increased focus on interoceptive stimuli, leading to increased tics [16].

The impact of external variables on tic frequency has long been noted. For instance, a 1978 longitudinal study of one patient noted that stressful life events overcame any positive medication effects from haloperidol [28]. In one study from 2003, 76 adults with either habit disorders or chronic tic disorders kept a daily diary of which activities were associated with tics [29]. While there were some trends in which types of activities were low or high risk for triggering tics, it was also evident that different people reacted differently to the same situations. For instance, 45% of patients with chronic tic disorders found passive attendance (i.e., watching TV or attending a hockey game) to be low-risk, while 38% found it to be high-risk. Overall, high-risk activities included socialization, waiting, and being in transit, while low-risk activities included studying and physical activity. People associated high-risk activities with tenseness, boredom, dissatisfaction, and disinterest. A Himle et al. study from 2014 showed similar results, with the most commonly noted antecedents for tics being TV or video games, home after school, homework, school classrooms, and public places [4]. This study also helped provide evidence that the consequences that occur after tics can help to explain setting effects. For instance, more attention after tics occurred 82% of the time at home after school as compared to only 37% of the time during sports/physical activity. This difference in the attention paid to tics may partially explain the common sentiment that tics are worsened at home after school versus during physical activity.

### 3.4. Stress

Nearly all patients report that stress worsens their tics [8]. However, analyzing the relationship between stress and tics is challenging, both because stress is multifactorial and because different people can react to the same stressor very differently depending on a variety of factors [16].

When measured in the laboratory setting, several studies have actually shown that tics decrease during stressful tasks. In one study of eight children with co-occurring tic and anxiety disorders, children’s heart rates and tic frequency were measured while completing stress-inducing tasks such as public speaking and discussion of family conflict [30]. They found no correlation between higher heart rate (indicating physiologic arousal) and tic frequency and, in fact, found that tic frequency decreased with higher heart rates during the speech task. In another study of 31 children and adolescents with tic disorders, tic frequency decreased while participants completed the Trier Social Stress Test, even though physiological markers of stress were present [31]. However, the techniques used to induce stress in these participants were task-based, and it is possible that the tasks themselves influenced the participants’ tics as they constituted a form of distraction.

While it is not clear that laboratory simulations of stress increase tic frequency, there is some evidence that this type of stress may impair a patient’s ability to suppress tics. In one study, 10 adolescents with TS attempted to suppress tics during stressful and non-stressful conditions [32]. Although there was no difference in tic frequency between the baseline stressful and non-stressful conditions, the participants were less able to suppress tics during stressful conditions. This result may explain the widespread perception that stress worsens tics. As far as treatments are concerned, relaxation therapy appears to be ineffective as monotherapy, again suggesting that stress per se plays only a limited role in the overall TS picture [33].

In general, laboratory conditions are best able to simulate short-term, task-based stressors, but many patients use the term “stress” to instead refer to stressful life events. In one 2003 study of children with TS and OCD, although children with these conditions scored higher than controls on a clinician-rated measure of psychosocial stressors, the Yale Children’s Global Stress Index (YCGSI), there was no correlation between the YCGSI score and tic severity [34]. However, there was a correlation between the scores on the self-reported Daily Life Stressor Scale (DLSS) and tic symptoms severity. The YCGSI was a clinician score measuring major life stressors during the past several months, while DLSS was a patient-scored measure of minor life stressors over the past week. These results suggest both that tic severity correlates better with daily life stressors than major life events and that patients are better able to evaluate their stress exposure than clinicians. This is perhaps because different patients can experience the same event as stressful or not stressful based on a wide range of individual factors.

Other studies show similar results. In a prospective longitudinal study from 2004, there was a small but statistically significant correlation between negative minor life events and tic severity on the aggregate level [35]. However, when the same study looked at the association on the individual level, only a minority of participants demonstrated a significant relationship between the two. A later prospective longitudinal study found that current psychosocial stress predicted future tic severity but not current tic severity [36]. A third study found a correlation between both major and minor life events and motor tic severity, but not vocal tic severity [37]. Taking all real-world studies together, it is clear that recent minor life stressors can play a significant role in increasing tic severity, despite the fact that laboratory-induced stressors have not been shown to have the same effect.

### 3.5. Rewards for Tic Suppression

It has been well established that rewarding patients for tic suppression decreases tics. A common study design to test this is the Tic Suppression Task, first developed by Woods and Himle [38]. In this task, the participants are placed in front of a computer or machine that they are told is a tic detector. A researcher observes unobtrusively, for instance, through a one-way mirror, to limit the impact of observation. The “tic detector” then rewards patients for tic-free intervals longer than 10 s, such as with poker chips that can be redeemed for money. The participants are observed during the baseline conditions, where they are instructed to tic freely, and suppression conditions, where they are instructed to suppress their tics and told they will be rewarded. Often, these conditions are measured several times in random order.

In one pooled analysis of nine studies using the Tic Suppression Task, 90% of the participants showed at least some suppressibility in response to immediately rewarded tic suppression [32,39,40,41,42,43,44,45,46,47]. Furthermore, about 70% of the participants showed at least a 50% reduction in tic frequency, and about 20% of the participants showed at least a 90% reduction in tic frequency. Factors associated with better suppressibility included more frequent tics and older age [48]. The addition of the stimulant dexmethylphenidate decreased tic frequency at baseline but, somewhat surprisingly, did not enhance tic suppression during the reward task [49].

Further research has demonstrated that the reward in the Tic Suppression Task must be contingent on tic suppression in order to be effective. A common research design includes a baseline condition, a verbal instructions condition, a Differential Reinforcement of Zero-rate Ticcing (Differential Reinforcement of Other behavior: DRO), and Noncontingent Reinforcement (NCR) [43,50]. The participants are instructed to suppress tics and receive rewards in both the DRO and NCR conditions. However, in the NCR conditions, the participants receive rewards regardless of the frequency of tics. The DRO condition outperforms both the NCR and verbal instruction conditions [43,50].

Importantly, it is possible to create a tic-reducing condition without the participants’ knowledge. In Woods et al. 2009, the participants completed the Tic Suppression Task with baseline, verbal instruction, and DRO conditions [51]. During the verbal instruction condition, an orange light was illuminated, and during the DRO condition, a purple light was illuminated. No lights were illuminated during the baseline conditions. The lights were not explained to the participants. The participants completed four different training sessions under these conditions. During a fifth session, the lights were illuminated during the corresponding tasks, but no suppression instructions or reinforcement were provided. Despite this, there was a statistically significant decrease in tics when the purple light was illuminated compared with no lights being illuminated. In other words, the participants had been trained without their knowledge to suppress tics when they observed a previously irrelevant stimulus, the purple light. This result has important implications as it can help explain tic variability between different environments, as there is likely differential reinforcement of tics between these different environments. This is the concept behind the functional intervention component of the Comprehensive Behavioral Intervention for Tics (CBIT), which has been tested in large, randomized controlled trials and is the main component of the CBIT-Jr intervention designed for younger children [52,53,54].

### 3.6. Distraction/Focused Attention

Multiple studies have demonstrated that tic severity worsens transiently while one focuses one’s attention on tics. One study of two children showed that talking about their tics increased tic frequency while talking about other topics did not [55]. A later study of 12 patients found that tic frequency increased when patients observed themselves ticcing via a mirror but decreased when watching a video of themselves without tics [56]. The impact of distraction, or focusing attention on something other than tics, is less consistent throughout different studies. Some studies have found that focusing attention away from tics reduces tic frequency [57]. Others have found that distraction interferes with tic suppression, thus increasing tic frequency during suppression conditions [58]. Still others have found that distraction does not impair tic suppression but that performance on the distraction task was worse during periods of tic suppression [42]. This suggests that tic suppression requires significant attention, which is perhaps why some studies show that distraction worsens patients’ ability to suppress their tics and supports patients’ observations that tic suppression can be burdensome and interfere with performance on school exams.

### 3.7. Being Observed

Studies differ in their findings as to how different types of observation by others affect tic severity. In one study that looked at tic severity at home and in the office as measured by video recordings, the patients had significantly increased tics at home [59]. The patients were recorded both with and without others in the room, and tic severity was increased when the patient was alone. However, in a similar study conducted by a different group, there was no difference in tic severity between recordings taken at home or in office, and patients had more tics when being overtly observed as compared to covertly observed [60]. Finally, a third study, which measured tic frequency across five different daily life activities, found that tic frequency was at its lowest when patients were alone [10]. Thus, the impacts of being observed are not clear.

### 3.8. Social Media Exposure

Anecdotal reports from clinicians indicate an increase in patients presenting with tics or tic-like symptoms since the COVID-19 pandemic began, and many attribute the increase to content labeled as TS on social media. Often, the symptoms on these social media sites do not represent classic Tourette’s syndrome symptomatology and instead are more consistent with a type of functional neurological disorder [61,62,63,64,65]. While there is limited research on how the popularity of such videos on social media has impacted those with pre-existing diagnoses of Tourette’s syndrome or other chronic tic disorders, it is possible that exposure to such content may increase tics in this population as well.

### 3.9. Social Situations

Several studies have found that tics are worsened in association with some social situations [4,37,66]. For example, one study showed that minor daily life events were associated with increased tics, and the most frequently cited daily life events were those involving relationships with friends [37]. A different study of patients with chronic tic disorders examined the attention paid by participants to words associated with social threat as compared to benign words [67]. They found that the CTD participants had increased attention to threat as compared to controls and that increased attention to a threat was associated with higher scores on some measures of tic severity, suggesting that social conflict and sensitivity to a social conflict may worsen tics.

### 3.10. Music

Musical performance has also been shown to decrease tics. One survey of 183 musicians with TS showed that on a scale of one (drastic symptom worsening) to five (drastic symptom improvement), the participants averaged a 4.45 rating for their symptoms while they performed music [68]. A different study videotaped eight musicians while performing, thinking about performing, and listening to music [69]. All three conditions decreased tics from baseline, with the largest improvement coming while performing. Additionally, there was a short-term decrease in tics in the period after they performed music, although the effect was small.

### 3.11. Exercise

Exercise has also been shown to decrease tic frequency, both during the exercise itself and to a lesser extent in the period immediately following the exercise [70,71]. One study also found a significant negative correlation between exercise and obsessive-compulsive behaviors, anxiety, and depression in children with tic disorders [71]. It is possible that the improvements in these other factors can partially explain the benefit in terms of tic improvement that is seen.

### 3.12. Diet and Supplements

Many families report that certain food types worsen tics. In one survey of 224 participants with TS regarding dietary habits and tic frequency, there was a significant negative effect on tics for foods containing caffeine, preserving agents, refined sugar, and sweeteners [72]. However, given that this study was based on participant self-reports, more study on this topic is necessary. The treatment of tics with the addition of certain foods or supplements has also been attempted, such as probiotics, N-acetyl cysteine, and omega-3 fatty acids, but results have generally not shown significant improvements [73,74,75]. One open-label study did show improvement in tics in patients given L-Theanine and Vitamin B6 as compared with a control condition of psychoeducation [76]. Additionally, an open-label study of magnesium and vitamin B6 showed an improvement in children with TS suffering from clinical exacerbations [77]. However, further placebo-controlled studies are necessary.

## 4. Relation to Brain Pathophysiologic Models of Tic Production

Various authors have proposed models of regional brain activity that might produce tics [16,78,79,80,81,82,83]. Godar and Bortolato most directly address the potential role of immediate environmental effects in their model [16], but most models do not attempt to make such a link. Few studies directly measure the effects of factors that transiently affect tic severity on regional pharmacology or neuronal activity. Therefore, the available evidence has limited ability to support any one pathophysiological model. Imaging or EEG technology could be used to identify the brain regions activated while exposed to specific antecedents or consequences. In the future, new techniques based on machine learning and computer modeling may help to investigate the explanatory power of specific models given such data.

## 5. What to Do about It

We can take advantage of these situational effects on tics to improve tic severity and frequency. In other words, if the environment can worsen or improve tic severity, we can exploit such effects to improve the lives of people with tics. To start, most experts recommend creating a “tic-neutral environment” [84]. This is intended to minimize the antecedents and consequences of tics, thus reducing tics. CBIT and CBIT-Jr are evidence-based treatment modalities for TS that work on this concept [52,53,54]. CBIT focuses primarily on habit reversal training, while CBIT-Jr is adapted for younger patients and focuses heavily on parent–child interactions. Both include a functional assessment and intervention, which is meant to identify and then modify tic antecedents and consequences. For instance, negative parental reactions may be identified as consequences of tics. As an intervention, parents would then be taught to minimize their reactions to tics [52]. The Tic Accommodation and Reaction Scale is a validated measure to identify environmental antecedents and consequences and could be used to track the effectiveness of these types of treatments [11].

Patients often resist tic-suppression-based behavioral interventions such as CBIT and exposure and response prevention (ERP) based on perceptions acquired over years of intermittent efforts to suppress tics. Occasional, spontaneous efforts at tic suppression can lead to frustration, which can paradoxically worsen tics or even reinforce ticcing. However, consistent, expert-directed tic suppression is an important part of these treatment modalities.

There is also evidence that treating anxiety disorders, as well as managing less severe anxiety, can be helpful. Relaxation biofeedback has been shown to be potentially helpful in the treatment of tics [85]. A small pilot study also showed positive results for a combination of neurofeedback combined with imagery training, which may induce a state similar to the state of focused attention, which is often thought to reduce tics [86]. Other common-sense suggestions have less of an evidence base, such as music therapy, managing fatigue, improving sleep, exercising, engaging in enjoyed activities, and distracting attention from tics. However, these are likely to be helpful as well, and many are good recommendations for all patients whether or not they have a chronic tic disorder.

## 6. Future Directions

As mentioned above, there are currently many factors for which evidence is mixed regarding their impact on tics. These factors, such as stress, distraction, and observation, would certainly benefit from further research. Other factors have limited research altogether, such as the impacts of social media, dietary changes, parenting styles, sibling roles, and educational environments. New research techniques could also improve the quality of the research on this topic. For example, machine learning, computational modeling, and imaging approaches could be useful in linking behavioral antecedents and consequences to underlying brain mechanisms. Other novel approaches, such as wearable technologies, could also be used to track tic frequency as a method of momentary ecological assessment in the future to help provide more accurate measurements of tics in response to real-world situations.

## 7. Summary

In summary, many factors can affect the moment-to-moment expression of tics (Table 1). Variables such as fatigue, anxiety, and certain types of thoughts have all been shown to worsen tics. The impact of other variables has been more variable. Different types of stress have been shown to both increase and decrease tics, and the impact of stress on tics continues to be an active area of research. Distraction and being observed similarly have had mixed impacts. Some areas of research, such as the impact of social media exposure and different foods, would benefit from more extensive research in the future. However, it is clear overall that the environment can significantly impact tic severity, even when the patient is not aware of the impact or is no longer receiving behavioral reinforcement. Additionally, reinforcement for tic suppression has been clearly shown to improve tics and has formed the basis for many common forms of treatment for TS. As our understanding of which factors worsen and improve tics continues to grow, the treatment options for TS should continue to improve as well.

## Figures and Tables

**Table 1 jcm-11-05930-t001:** Summary of factors that impact tics.

Improves Tics	Mixed Effects	Worsens Tics	Unclear Effects
Rewards for tic suppression Musical performance Exercise	Stress Distraction Observation by others	Fatigue Anxiety Thinking about tics Attention to tics Social conflict	Social media Some foods Dietary supplements

## Data Availability

Not applicable.

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
