# Peer review of "Why Tic Severity Changes from Then to Now and from Here to There"

_jcm, 2022, doi:10.3390/jcm11195930_

Round 1

Reviewer 1 Report

First, I would like to thank the editor for the opportunity to review the manuscript titled Why tic severity changes from then to now and from here to there. The authors provide a brief overview of the various internal and external factors that can affect tic severity. I think the review is can be a helpful addition to the literature. I only have a few comments that are listed below:

On page 3 the following reference is made: "Other research indicates that tics are worsened with multitasking [27]." What type of multitasking? Any further information about this could be helpful to the reader.

Some of the statements in the section about stress and tics seem to be contradictory. At one time the following is stated: "While it is not clear that stress increases tic frequency, there is some evidence that stress may impair patient’s ability to suppress tics.", while later it says: "Taking all studies together, it is clear that recent minor life stressors can play a significant role in increasing tic severity." Please clarify. 

Author Response

We thank this reviewer for their thoughtful comments. Please see the attachment below.

Reviewer 2 Report

In my opinion, the topic of the paper is of some interest. However, there are some critical issues that the authors have to address before publication.

1) The paper focuses on the roles played by, for example, environmental and behavioral aspects of tics production. In this respect, an important lack of the paper is that the authors do not make any discussion about possible brain mechanisms underlying tics production. To really increase the understanding of the “spectrum” of tics disorders and the relationship between proximal and distal causes supporting tics production is fundamental to discuss recent evidence and theories about brain mechanisms underlying tics production. The authors should try to make an explicit link between those mechanisms and the factors influencing tics already discussed in the paper (e.g., stress, food, etc.). The paper could be really improved by adding this part. To add this part could be critical to discuss the system-level approach to tics in Tourette proposed in the following paper:

Caligiore, D., Mannella, F., Arbib, M. A., & Baldassarre, G. (2017). Dysfunctions of the basal ganglia-cerebellar-thalamo-cortical system produce motor tics in Tourette syndrome. PLoS computational biology, 13(3), e1005395. 

2) The authors should add a table to summarize the positive or negative influence of several factors on tics production. The reader will certainly benefit from this table.

3) The authors should add a figure to make explicit the relationships between several factors influencing tics production and severity (environmental, lifestyle, etc.) and possible system-level brain mechanisms underlying tics production.

4) Finally, the authors could briefly mention in the conclusions the benefit of using new techniques based on machine learning and computational modeling tools to investigate through computer simulations the relationship between different factors producing tics.

Author Response

(The authors gave the same response as above.)

Reviewer 3 Report

The authors have summarized the literature on internal and external variable associated with tic exacerbation and attenuation in Tourette’s disorder. The authors have synthesized this literature well, especially given the heterogeneity in methodological approaches. I like that the authors highlight that a challenge in studying antecedent and consequence variables is essentially that individuals are exposed to multiple variables at a given time and there may be interindividual intraindividual variation in the experience of these triggers due to the intersection of external and internal factors. It is highly nuanced. Below I have listed minor suggestions for improvements.

There is a subheading that is repeated, once without a typo and once with a typo: “Just the facts, ma’am.”

Minor points, but I would suggest referring to relaxation training rather than relaxation therapy at p. 4, line 184; and it should be “tic suppression task” rather than “tic suppression test” just to keep terms consistent across literature.  

The authors mentioned supplements for tics on p. 7, second paragraph. What are their thoughts on magnesium? It could be worth mentioning here.

In the “What to do about it” section, the authors refer to creating a tic neutral environment to minimize the impact of the environment on ticcing, and state that CBIT work on this concept. Can the authors elaborate on this by describing the function-based assessment and intervention component of CBIT with some example.

Another point that could be worth mentioning in the “What to do about it” section is the potential utility of training individuals with tics to achieve the same mental state experienced when performing music or another similar activity associated with tic attenuation. I’ve heard people refer to a “flow state” to describe what might be going on. I wonder if mindfulness could be used to achieve this. Feel free to disregard if this point overlaps too much with neurofeedback.

The authors state that tic suppression is an important part of the treatment modalities above. Can this sentence be rephrased to make sure it is known that tic suppression is part of ERP for tics and not CBIT? It seems that CBIT providers tend to emphasize the distinction between use of competing responses in CBIT, and suppression of tics.

An area of future research that might be worth mentioning is the use of imaging or EEG to identify the brain regions/brain waves activated while engaged in/exposed to specific antecedents or consequences. The authors discuss interventions but do not discuss how novel assessment approaches could enhance our understanding of the influence of antecedent and consequence factors on ticcing, and how these might guide future treatments. 

Author Response

(The authors gave the same response as above.)

Reviewer 4 Report

Dear Editor of the Journal of Clinical Medicine

Thank you for giving me the opportunity to review this study. The authors have written the text so that it can be understood and used by anyone, even non-specialists; I consider this the study's strength. However, there are some criticisms of the current review which I think need to revise for publication.

The authors seem to be looking to identify variables that affect the occurrence of tic disorder. Therefore, they have investigated these variables and classified them. They classified identified factors as external and internal variables.

The authors mentioned sleep, anxiety, and thoughts as internal variables and distraction/focused attention under the title of external variables. Anxiety is a co-occurring disorder for tic disorder and it cannot be easily named as an internal variable. We don’t know whether tic is a primary diagnosis followed by anxiety or vice versa. This is also true of other comorbid disorders, for example, sleep disorders, attention disorders, or other neurodevelopmental or psychiatric disorders.

In the main text of the manuscript (line 140) “Other research indicates that tics are worsened with multitasking.” It may refer to weakness in functional executive performance and organizational skills. It doesn’t seem thought. It seems as a cognitive variable that may refer to some cognitive impairment that we observe, for example in ADHD.

Distraction/focused attention (line 250) seems noticeable to ADHD as a comorbid disorder and it’s not an external variable.

The authors classified stress as an external variable, it seems they talk about stressful situations (events). Otherwise, the experience of stress is internal. In fact, the evidence provided by the authors points more than anything to the weakness of these children in coping with stress.

In the following, rewards for tic suppression are a type of coping way to manage tic symptoms, it’s not an external variable same as other variables. However, here are debating whether this desire to suppress the tic could increase or decrease the symptoms or what is the contribution of other confounding variables.

Other variables mentioned by the authors in the title of “miscellaneous” can be classified under the title of external variables. for example, social situations (289), learning situations (music performance line 298), Exercise (line 306), and food types (line 312).

Some important variables were ignored in the current study for example role of family, parenting style, sibling role, the role of friends or educational environment, and so on.

If the authors are interested in revising their manuscript, I have a possible suggestion. They could divide all influential variables to decreasing or increasing tic disorder into two categories: personal variables including co-occurring disorders, emotional problems, cognitive impairments, and so on, and environmental variables which can include stressful events, family and parenting variables, social variables friends, school’s role, etc. Then the authors can discuss how this set of internal and external variables in interaction with each other can reduce or increase the tic.

Author Response

(The authors gave the same response as above.)

Round 2

Reviewer 2 Report

The authors properly addressed the raised concerns. In my opinion, the paper could be published now.